# Validation of a Low-Cost Pavement Monitoring Inertial-Based System for Urban Road Networks

**DOI:** 10.3390/s21093127

**Published:** 2021-04-30

**Authors:** Giuseppe Loprencipe, Flavio Guilherme Vaz de Almeida Filho, Rafael Henrique de Oliveira, Salvatore Bruno

**Affiliations:** 1Department of Civil, Constructional and Environmental Engineering, Sapienza University, Via Eudossiana, 18-00184 Rome, Italy; salvatore.bruno@uniroma1.it; 2Department of Transportation Engineering, Polytechnic School of the University of São Paulo, Avenida Professor Almeida Prado, Travessa 2, 83-05508010 São Paulo, Brazil; flaviovaz@usp.br (F.G.V.d.A.F.); rafa.oliveira@usp.br (R.H.d.O.)

**Keywords:** pavement monitoring, inertial measurement unit, urban road, international roughness index, ride number, ride comfort

## Abstract

Road networks are monitored to evaluate their decay level and the performances regarding ride comfort, vehicle rolling noise, fuel consumption, etc. In this study, a novel inertial sensor-based system is proposed using a low-cost inertial measurement unit (IMU) and a global positioning system (GPS) module, which are connected to a Raspberry Pi Zero W board and embedded inside a vehicle to indirectly monitor the road condition. To assess the level of pavement decay, the comfort index *a*_wz_ defined by the ISO 2631 standard was used. Considering 21 km of roads with different levels of pavement decay, validation measurements were performed using the novel sensor, a high performance inertial based navigation sensor, and a road surface profiler. Therefore, comparisons between *a*_wz_ determined with accelerations measured on the two different inertial sensors are made; in addition, also correlations between *a*_wz_, and typical pavement indicators such as international roughness index, and ride number were also performed. The results showed very good correlations between the *a*_wz_ values calculated with the two inertial devices (R^2^ = 0.98). In addition, the correlations between *a*_wz_ values and the typical pavement indices showed promising results (R^2^ = 0.83–0.90). The proposed sensor may be assumed as a reliable and easy-to-install method to assess the pavement conditions in urban road networks, since the use of traditional systems is difficult and/or expensive.

## 1. Introduction

The management of public infrastructure assets is a complex process that integrates many multidisciplinary strategies for their maintenance [1]. Generally, the process focuses on the later phases of the infrastructure life-cycle, but it would be better to integrate this process into the design phase [2].

This process aims to organize and implement strategies to maintain infrastructures and extend their life span, enhancing their performance [3]. In fact, the infrastructures and in particular the transport ones are fundamental components for maintaining the quality of life in society and the efficiency of the Countries’ economy.

Road pavement is a very important transport infrastructure asset that requires an accurate assessment of the distresses for understanding how to fix them. Pavement management systems (PMS) have been employed by road agencies in North America since the 1970s to manage their networks; these systems have evolved over the years to become reliable tools for the effective management of pavements for all road networks; since then their use has spread to all countries of the world [4].

Pavement distresses, causing surface unevenness, affect the vehicle operating cost [5], speed [6], riding comfort [7], safety [8], fuel consumption [9], wear of tires [10], noise [11] and pavement service life [12]. In addition to the direct surface monitoring (by visual or automatic inspection) of appropriately categorized distresses, pavement assessment [13] can take into account, roughness and/or ride evaluation [14].

The pavement roughness can be measured directly using high-performance equipment (contact or non-contact profilers), which detect road profiles along the pavement [15], and the acquired data are evaluated in terms of globally recognized indices worldwide [16].

The most popular index used around the world to evaluate pavement roughness starting from the measured profile is the International Roughness Index (IRI) [17]. Many threshold values are available depending on the profile length, the type of pavement (asphalt concrete or Portland cement concrete), and other pavement characteristics [18,19,20,21]; other interesting researches have proposed different threshold values considering the operating speed of the road [22,23], so as to accept higher IRI thresholds for the roads where the operating speed is lower. The costs associated with sophisticated pavement evaluation equipment such as a road surface profiler (RSP) can be significant [24,25] relative to the low budgets of road agencies. For these reasons, the RSPs are currently used to evaluate pavement roughness, as a component of a more complex road asset collection system (RACS), in nonurban road networks (roads outside the administrative borders of cities with speed limits of more than 70–80 km/h).

There are also operative problems that limit the use of RSPs in urban road networks: these devices provide reliable results only at certain measurement speeds, generally higher than 30–35 km/h, which could be unfeasible in urban areas for various reasons (the presence of speed limits, the low horizontal curve radii, the numerous intersections, etc.). In addition, these non-contact profilers need an obstacle free a launch segment that allows them to reach a predetermined survey speed, which further limits their application in an urban context. It should also be considered that the medium level of distress of urban pavements often prevents the correct operation of these systems [14], which, as mentioned, are designed for nonurban roads.

Alternative methods can evaluate indirectly pavement roughness considering ride quality indices; these indices thus defined can be determined starting from pavement profiles (i.e., ride number, RN) or considering methods, that involve the use of an accelerometer mounted in a moving vehicle. These last methods are potentially useful tools for pavement condition assessment in a cost-efficient way, but a preliminary calibration could be required to take into account the dynamic characteristics of the test vehicle and its speed [14].

Whatever system is used to evaluate pavement roughness (through direct or indirect methods), this should be integrated at least with a high-precision global positioning system (GPS) receiver to allow a correct localization and positioning of the measurements on the road [26,27,28,29,30,31,32]. The essential measurement systems necessary for the ride evaluation (three-axial accelerometer and GPS module), are already available in the modern smartphones where they are suitably integrated and synchronized [33,34,35,36,37,38,39]. For this reason, smartphones have been recently proposed to evaluate road condition over the world using apps with different approaches. Some apps try to estimate IRI along the surveyed road (which is divided into constant segments, 20–50–100 m) to provide a typical evaluation of pavement quality [40]. Other apps propose new indices [41] or categorize the acceleration peaks [42] in order to evaluate pavement conditions.

The urban pavements present also other management problems that consist in the inadequacy of thresholds for the roughness indicators currently in use considering the operating speeds, generally below 50 km/h [22,43,44]. Where an attempt has been made to overcome this shortcoming, such as some limits of the IRI defined according to the type of road or pavement, there are still some doubts about their applicability in urban road networks [45,46].

It is with all these considerations in mind that the choice of the monitoring system and the assessment method for urban pavements could be overcome by using an index that depends on the vertical accelerations measured inside a moving vehicle, taking into account its characteristics [47,48,49,50].

The pavement evaluation systems based on the survey with a low-cost device of the vertical accelerations inside the passenger compartment seem to be an interesting alternative to solve the difficulties in the monitoring and assessment of urban pavement.

These systems could arouse interest for those road network managers who do not yet have any continuous monitoring system for their pavements. Generally, they choose maintenance strategies and the related interventions to be taken regardless of any monitoring activity (time-based maintenance) or in consequence of failures (run-to-failure maintenance) with serious losses in terms of direct and indirect costs for the community. Instead, it would be useful to carry out maintenance referring to performance-based systems that allow identifying the appropriate time to perform maintenance interventions with respect to the conditions of the entire network and the available budget.

In this study, a novel inertial-sensor based system is proposed by using a low-cost inertial measurement unit (IMU) and a GPS module, which have been both connected to a Raspberry Pi Zero W board [51] and embedded inside a vehicle for monitoring the road condition indirectly. To assess the level of pavement decay, the comfort index *a*_wz_ defined by the ISO2631 [52] standard was considered.

Considering 21 km of roads with different levels of pavement decay, validation measurements taken with the proposed sensor, another pre-assembled high performance inertial sensor (Landmark 10 GPSA-150-10-200, Gladiator Technologies 8022 Bracken Pl SE Snoqualmie, WA 98065, USA), and a Class I inertial RSP [53] were performed. Therefore, comparisons between *a*_wz_ determined with accelerations measured on the two inertial sensors are made; in addition, correlations between *a*_wz_, IRI and RN which have been determined using respectively the inertial sensors and the RSP, were also performed. The results showed very good correlations between the *a*_wz_ values calculated with the novel sensor and the ones obtained using the reference pre-assembled sensor. Besides, the correlations between *a*_wz_, IRI and RN showed promising results. The proposed sensor may be assumed as a reliable and easy-to-install method to assess the pavement conditions in urban road networks, since the use of traditional systems is difficult and/or expensive.

## 2. Pavement Monitoring Sensors

In this section, the systems and instruments used in this paper for monitoring pavement conditions are described.

### 2.1. General Architecture of the Proposed Sensor

The proposed low-cost and easy-to-operate device has as main aspect its similarity with smartphones regarding sensors configuration, performance, and cost. Thus, the two devices assembled and set up for the described work are composed of the following consumer-grade components: a Raspberry single board microcomputer, a micro-electrical mechanical IMU, a mini GPS module, a power supply, and a flashcard. This section describes these components as follows and highlights the most important features regarding the described application.

#### 2.1.1. Raspberry Pi Zero W Single-Board Microcomputer

The Raspberry Pi Zero W is a low-cost single-board microcomputer of 6.5 × 3.0 cm developed by the Raspberry Pi Foundation for applications such as education and prototyping. This Raspberry model has a 512 RAM, a 1 GHz single-core microprocessor, and a 40-pin general-purpose input/output (GPIO) [51]. It also has 802.11 wireless LAN (Wi-Fi) and Bluetooth connectivity, which simplifies remote control and data transmission without the need for physical uninstallation and reinstallation. The Raspberries used in the described tests run the Linux-based Raspbian operating system.

#### 2.1.2. Inertial Measurement Unit

A micro electro-mechanical system (MEMS)-based IMU is a single chip multi-axis sensor that estimates at least linear accelerations and angular velocities and, thus, integrates accelerometer and gyroscope. Some versions of MEMS IMU single chips also integrate non-inertial sensors such as magnetometer and barometer. The recent MEMS technology progress focused on mobile gadgets has been yielding very low cost and very small smartphone-grade IMU units with a cost of about cents, size of about square centimetres, and satisfactory performance for non-critical applications. Thus, the main advantages of these inertial sensors when compared with traditional mechanical and solid-state sensors are the size reduction, the low power consumption and the low production cost [54].

For this research, we used the InvenSense MPU-9250 (TDK InvenSense, San Jose, California 95110 CA, USA), a 10 degrees-of-freedom module of 1.4 × 1.4 cm size. This inertial module integrates the three-axial MEMS inertial sensors (accelerometer and gyroscope) to a magnetometer and a BMP280 pressure module (a barometer plus a thermometer) [55,56]. The voltage readings from the inertial sensors are digitized using on-chip 16-bit resolution analog-to-digital converters (ADC) for each axis, and this digital output is sent to the Raspberry through an inter-integrated circuit (I2C) interface. Besides the raw measurements, the MPU-9250 module measures and has a digital motion processor that provides fused output for gesture recognition applications. Table 1 presents the main features of MPU-9250 accelerometer, gyroscope, and magnetometer.

The C++/Python library named RTIMULib [57] was used for sensors setup, initial calibration on Raspbian, and conversion of values from hexadecimal to floating-point representation. The following data is obtained:(1)three-axial raw linear accelerations (including gravity) in the sensor frame, in g;(2)three-axial raw angular velocities in the sensor frame, in rad/s;(3)three-axial raw magnetic field in the sensor frame, in µT;(4)pressure, in hPa;(5)height derived from the barometric calculation, in m;(6)temperature, in °C;(7)sensor attitude (roll, pitch, and yaw, in degrees).

Regarding attitude data, RTIMULib through an extended Kalman filter (EFK) obtains angles integrating inertial and magnetic data, a technique that adapts Kalman filter to a nonlinear problem such as the attitude estimation.

The sample rate was set up at 100 Hz given the optimum performance on preliminary tests, the aimed data analyses, and the usual sample rate for medium-grade smartphones. However, the maximum mean sample rate effectively obtained during operation (>10 s) was about 83 Hz owing to hardware and software limitations.

#### 2.1.3. Mini Global Positioning System (GPS) Module

A U-blox mini GPS module, NEO-6M model [58], was used in each sensor set. This receiver performs single-point positioning using C/A Code of L1 frequency from GPS constellation, as well as obtains augmented from satellite-based augmentation systems. The single-point positioning technique using single L1 frequency and civilian code presents a typical horizontal position error of 13 m at a probability level of 95% under standard scenarios. Complementary, satellite-based augmentation systems (SBAS) corrections reduce satellite-related and ionospheric-related errors and improve integrity, availability, and continuity. However, satellite-based positioning quality can be degraded by environmental factors such as signal multipath, signal blockage, and atmospheric interference [59,60]. NEO-6M main features are presented in Table 2.

The update rate for the GPS module was set up at 1 Hz regarding the performance during preliminary tests, and the sample rate for medium-grade smartphones. The lower sample rate in comparison with IMU rate requires interpolation of position, velocity, and time (PVT) data using the OS timestamp as the key attribute. Furthermore, GPS and IMU data are recorded in separated files since the sensor has the best performance under the abovementioned configuration. The Python library called GPSD [61] allows the acquisition, on the Raspbian environment, of PVT data through US National Marine Electronics Association (NMEA) protocol. The following GPS data has been obtained: geographic coordinates (latitude and longitude) of the acquisition point referred to WGS84 datum (GPS datum), geometric height, UTC time of the acquisition point, velocity, number of visible satellites, and uncertainty-related parameters.

Figure 1 shows the core components already assembled. The IMU module and the GPS module were connected to the processing unit and glued to the Raspberry case.

#### 2.1.4. Other Components

Each Raspberry operates with a 16 GB micro-SD used to store the operating system and the gathered data. Moreover, a portable rechargeable battery unit supplies power with 10,400 mAh capacity through a micro-USB port. Considering storage and power capacities under the aforementioned configuration, the sensor sets presented an autonomy of at least 50 h during the preliminary test.

### 2.2. LandMark 10 GPSA-150-10-200

In order to validate the results of the measurements made with the proposed sensor on the same test road test also a reference pre-assembled inertial platform LandMark 10 GPSA-150-10-200 [62] was employed.

The most important product characteristics are summarized in the code name, that report the operating range of both gyroscopes (±150°/s) and accelerometers (±10 g’s) as well as the product type GPS/Attitude Heading Reference System (AHRS).

The main components of this instrument are:the IMU (Figure 2a);the integrated GPS receiver (Figure 2b);the power supply to connect to a laptop (Figure 2c).

This connection also allows recording the data measured to a comma-separated values (CSV) file. The software named GLAMR [63] to acquire the data have to be installed in a standard Notebook.

The Kalman filter is automatically implemented inside the LandMark 10 GPSA-150-10-200; the Kalman filter is an efficient recursive filter that evaluates the state of a dynamic system starting from a series of measurements subject to noise. Its use eliminates part of the background noise that could affect the measurements.

### 2.3. Road Asset Collection System

In order to validate the results of the measurements made with the proposed sensor, a Class I inertial RSP was employed on the same test roads. The RSP is one of the main components of a road asset collection system (RACS) employed by the road managers to collect data on the road. In this research, the Laboratory of Road Materials and Maintenance of the Italian National Road Agency (Centro Sperimentale Strade di Cesano di ANAS S.p.A. [64] Gruppo Ferrovie dello Stato Italiane [65]) allowed us to survey some road sections with its RACS and they were shared the RSP data collected. The RACS collect asset data about objects, features, structures, and landmarks located along the Italian highways and road networks managed by ANAS for roads planning, management, and maintenance. It creates classified inventories annotated with object dimensions, object position relative to the road, and global position reference.

This RACS called “Cartesio” was designed according to the Department of Road Maintenance of the Italian National Roads Department (ANAS). The system has been in operation since 2018.

The main components of this RACS are:the positioning and orientation system (GPS and wheel odometer) so as to georeferenced the data collected with the other on-board sensors;on-board sensors (digital camera, DC) for inspection road asset and pavement, n. 5; light detection and ranging (LIDAR) to map roadside equipment and features, n. 2; a laser crack measurement system (LCMS) for automatic inspection of the pavement condition; RSP to collect longitudinal profiles;the synchronization system coordinated by a management system.

Other auxiliary systems are:the data storage system;the power supply system for equipment and documents.

All the components are permanently installed on a Fiat Ducato 290 vehicle. In Figure 3, some of the main components of Cartesio are depicted.

## 3. Pavement Evaluation Methods

In this section, the methods and procedures used in this paper for evaluating pavement condition are described.

### 3.1. Whole-Body Vibration—ISO 2631

Starting from the vertical accelerations in the time domain, measured by inertial sensors onboard the test vehicle, the root mean square (RMS) accelerations through the evaluation of the PSD can be determined for all the frequency range of interest for the human response to vibrations (between 0.5–80 Hz), and analyzed by a spectrum of 23 one-third octaves bands. This procedure is specified by the technical standards currently in use [66,67], and it is similar to other analysis to transform the signals measured in the time domain into spectrum in the frequency domain.

Once the RMS acceleration one-third octave spectrum $a_{z}=\left(a_{z,1},a_{z,1},\ldots ,a_{z,23}\right)$ corresponding to the 23 frequencies proposed by ISO2631 (0.5, 0.63, 0.8, 1, 1.25, 1.6, 2, 2.5, 3.2, 4, 5, 6.3, 8, 10, 12.5, 16, 20, 25, 32, 40, 50, 64, 80) is known, it is possible to calculate the vertical weighted RMS acceleration ($a_{wz}$) using Equation (1): (1)$$a_{wz}=\sqrt{\overset{23}{\underset{i=1}{\sum }}{\left(W_{k,i}\cdot a_{z,i}\right)}^{2}}$$
where $W_{k,i}$ is the *i*-th frequency weighting in one-third octaves bands for the sensor, provided by the standards ISO2631 [52] and $a_{z,i}$ is the vertical RMS acceleration for the i-th one-third octave band. Then, the calculated values can be compared with the threshold values proposed by ISO 2631 for public transport (Table 3), in order to identify the comfort level perceived by users in all road sections, also considering several running speeds.

Considering the real characteristics of the inertial sensor used during the measure and analysis (the analysis time, T and the sampling frequency, $f_{s}$), not all the 23 one-third octaves bands could be determined. At any rate, the evaluation of PSD was done using a DFT function in MatLab^®^ and considering the Nyquist–Shannon sampling theorem [68]. In addition, to minimize the effects of performing DFT over a no integer number of cycles, the classic technique of split-cosine bell windowing was used. In Figure 4, examples of three different spectra calculated from acceleration data are depicted.

In the Figure 4, it is stressed that the contribution in the *a_wz_* calculus of the last 4 values of the spectrum can be neglected, in consequence of the low values of the frequency weightings curve *Wz*. 

The ISO2631 was developed by the International Organization for Standardization (ISO) and it is regarded as a standard model adopted by several countries over the world. This standard provides several comfort levels (Table 3), introducing an overlapping zone between two adjacent ones because many factors (e.g., user age, acoustic noise, temperature, etc.) contribute to determine the degree to which discomfort could be noted or tolerated.

At any rate, the comfort levels proposed by ISO 2631 are adopted in many countries, and they may be compared with the RMS values of the frequency-weighted vertical acceleration in the vehicle *a_wz_* obtained inside a vehicle, giving approximate indications of likely reactions to various magnitudes of overall vibration total values in public transport.

In order to define specific limits to be used by road agencies, it is necessary to link *a*_wz_ values to IRI ones as proposed by many researchers [7,21,43,45,49,50]. In this way, it is possible to relate comfort perception (also influenced by vehicle characteristics) with a parameter that represents the condition and performance of road surfaces.

### 3.2. International Roughness Index (IRI)—ASTM E 1926

The IRI was elaborated by a World Bank study in the 1980s [69] and it is one of the most adopted indices used to evaluate the pavement roughness. It is based on a mathematical model called quarter-car and was developed in order to assess the pavement condition relating to all the detrimental effects such as ride quality, dynamic load increase, tyre rolling noise, fuel consumption, and road safety.

Many decay curves have been proposed to predict the maintenance plan over time [70] and the consequent service life of the pavement knowing its operating conditions (traffic, climate etc.) [71].

The calculation of IRI was performed using a computer program that implements the simulation of the mechanical model considering a profile according to Equation (2):(2)$$IRI=\frac{1}{L}\int_{0}^{L}\left|z_{s}-z_{u}\right|dx$$
where *L* is the length of the profile in km, *V* is the simulated speed set to 80 km/h, $z_{s}$ is the vertical displacement of the sprung mass in m, and $z_{u}$ is the vertical displacement of the unsprung mass in m. The final value is expressed in slope units (e.g., m/km or mm/m). In the present work, the algorithm proposed by the ASTM E1926 standard [72] for IRI calculation was used.

As reported in [44], there is a high heterogeneity of IRI thresholds adopted around the world. In fact, IRI limit values mainly depend on several aspects: road surface type (i.e., asphalt or cement concrete pavements), road functional category, average annual daily traffic (AADT), legal speed limit and segment length considered for IRI calculation.

The most common segment length indicated in non-US countries is equal to 100 m [44], but frequently also lengths of 50 m and 20 m are adopted to better take into account the contribution of the single event bumps with respect to the distributed unevenness.

Cartesio collected IRI every 10 m of road section adopting a profile length of 20 m.

### 3.3. Ride Number RN

The ride number (RN) is the result of a mathematical algorithm obtained using two longitudinal profiles that allows the estimation of the subjective ride quality perceived by road users. It is quite used over the world and it is correlated to the perceived comfort experimented by user riding on pavement roughness.

The RN index is the result of an international research conducted in the 1980s and sponsored by The National Cooperative Highway Research Program (NCHRP), with the aim of analyzing how the characteristics of road profiles influence the ride comfort perceived by road users [73].

The RN thresholds were obtained relating the road profile characteristics to the opinions of interviewed users about the roads; the pavement condition is defined on a 0-to-5 scale where 0 corresponds to “impassable” pavement condition and 5 to “perfect” one (Table 4).

The RN calculus requires a pavement survey using a “Class I” profiler of two profiles, and two Profile Indices (*PI_Left_*, *PI_Right_*) were calculated adopting the algorithm reported in the ASTM E 1489–98 [74].

The calculation of *RN* was performed by means of Equations (3) and (4):(3)$$PI=\sqrt{\frac{PI_{\;Left}^{2}+PI_{\;Right}^{2}}{2}}$$
(4)$$RN=5\cdot e^{-160\cdot \left(PI\right)}$$

With some exceptions, the wavelengths’ range of interest for RN is similar to that of IRI, as reported in some researches [17]; in consequence, good correlations can be found between IRI and RN [12]. In particular, RN presents a higher sensitivity to low wavelengths than IRI, which has a greater sensitivity to wavelengths of 16 m or longer than RN. 

## 4. Field Tests for System Validation

With the aim to validate the novel sensor, some field tests were carried out using two identical prototypes of the device described in Section 2.1. In this paper, these two sensors can be distinguished with the code “SENSOR#1” and “SENSOR#2”.

The accelerometer data recorded by the sensors placed inside test vehicles were processed using ad hoc program code written in MATLAB^®^ in order to calculate *a**_wz_* index values per second. For the acceleration signals, an analysis time of 2 s was considered, so, for each device, an overlap of 1 s the acceleration signals was obtained.

The validation test was performed identifying a total of about 21 km of roads (Figure 5) with flexible pavement located in the northern outskirts of Rome. The route started and ended at the same section; it was articulated on both urban and nonurban roads (Table 5) with one lane for each travel direction. 

Field tests were carried out without closing roads to traffic and no change in driving behavior was requested to the drivers of the test vehicles in which the sensors were placed, so speed value recorded during the measurements were variable in consequence to the road and traffic condition (Figure 6).

In order to have a correct interpretation of the results obtained from the proposed prototypes SENSOR#1 and SENSOR#2 during the measurement campaigns also additional instruments were used:LandMark 10 GPSA-150-10-200, a precision measuring instrument [75] with sampling frequency equal to 100 Hz. Post-processing acceleration data recorded from this IMU was aimed to obtain the frequency-weighted vertical acceleration *a_wz_* considering analysis times by one second each;“Cartesio” road asset collection system using three RSPs (PaveProf V2.0, PaveTesting, Letchworth Garden City Hertfordshire, UK [76]) able to measure the road profiles in the left and right wheel paths as well as in the center lane (Figure 7).

Information about the vehicles chosen for road tests and the IMUs’ position inside these test vehicles have been summarized in Table 6. The SENSOR#1 was positioned on the passenger side floor of a Renault Zoe together with the LandMark, while the SENSOR#2 was installed on the “Cartesio” dashboard.

No particular details (foams, rubber layers or similar) were adopted to fix the device parts to the box support (only screws, bolts and rubber bands) or to the vehicle dashboard (only double-sided tape), because it is foreseen in the future that these instruments should be able to be simply mounted without special provisions.

**Table 6 sensors-21-03127-t006:** Characteristics of the IMUs’ position inside the vehicles during the tests.

Test Vehicle	Average Test Speed (km/h)	IMU	IMU Position Inside the Test Vehicle
Renault Zoe ^1^	44	SENSOR#1	Passenger side floor (Figure 8a)
		LandMark	Passenger side floor (Figure 8a)
Cartesio	37	SENSOR#2	Dashboard (Figure 8b)

^1^ Full electric, Car production year: 2020, Mileage: 190 km.

## 5. Results and Discussion

A first comparison between the low-cost pavement monitoring SENSOR#1 and the LandMark has been carried out in terms of speed values collected by GPS modules during the survey by both devices at the same time (Figure 9). In the few points where the speed values are not matched between the two sensors, a log error of “went out of sync” were registered on Landmark due probably to a loss of signal. The management software of the sensor corrected these speed values using Kalman filter in order to obtain measurements that are more efficient. The two devices collected speed values with different frequency rate: 100 Hz for the LandMark and 1 Hz for the proposed SENSOR#1.

The comparison between speed values collected by the two devices at the same time sample showed a good correlation (Figure 10).

In some isolated positions, a maximum difference of 20% was registered, and, in total, an average total value of only 0.2% between the two speed values was obtained.

After this preliminary comparison, as a result of data processing with reference to the examined roads, numerical values of IRI and RN yielded by “Cartesio” every 10 m long the road, and the whole vibration index *a*_wz_ every 1 s based on accelerometer signal (analysis time 2 s) measured using different IMUs were obtained.

Considering that the vehicle speed in the IMU device tests was in the range of 10–16 m/s, it was possible to determine the measurements of the aforementioned indices every 1 s and, therefore, approximately every 10–16 m. It was not considered useful, as well as difficult, to exactly match the position in which all the indices (*a_wz_* obtained with 3 devices, IRI and RN) were available (Figure 11). For this reason, in this preliminary validation phase, fixed and constant long road sub-sections (100 m) were considered. The average value of the indices that the positions were included in a generic section were assumed representative for that sub-section.

For operational reasons related to the use of the manager’s profiler, the considered 21 km total section of urban and nonurban road network was measured during the morning of a working day with traffic conditions that did not always allow the vehicle to move at the minimum speed for the correct survey of IRI and RN measures.

Consequently, not all the collected measurements were considered in the validation.

In the 100 m sub-sections where the speed of the vehicle used for pavement profile collection was greater than the minimum acceptable value to have reliable indices, a subdivision into performance classes with reference to the pavement decay was adopted.

Three different pavement condition categories (“Good”, “Fair”, and “Poor”) derived from related researches [22,43,47,77] were adopted considering the IRI threshold values (Table 7).

For this validation phase, it has been assumed to consider road sections where the pavement conditions did not vary continuously from the previous to the following 100 m sub-section. On the other hand, the variability of pavement conditions is quite frequent during a normal survey regardless of whatever index is adopted. For this reason, in the usual practice of the pavement monitoring procedure, it has been necessary to identify appropriate homogeneous sections in relation to the deterioration conditions surveyed [78].

On the contrary, during the validation procedure, in the entire 21 km road section, 3 sufficiently long sections (at least equal to 400 m) respectively in good, fair and poor conditions were identified (Figure 12 and Table 8).

### 5.1. Comparison between SENSOR#1-a_wz_ and LandMark-a_wz_

The first step in results analysis was to find a relationship between *a*_wz_ values calculated from data collected by SENSOR#1 and *a*_wz_ values based on data collected by LandMark 10 GPSA-150-10-200 (Figure 13). For clarification purposes, it is important to underline that both the devices, one next to the other, were inside the same vehicle during the same test.

The regression results showed very good correlations between the frequency-weighted vertical accelerations calculated with the proposed device (SENSOR#1) and the ones of the reference sensor (LANDMARK). The coefficient of determination (R^2^ = 0.98) indicates a strong relation between the measurements from the SENSOR#1 and the LandMark10, although the first presented a smaller accuracy, and it provided *a_wz_* index values about 12% greater than the ones of the reference IMU.

### 5.2. Comparison between SENSOR#1-a_wz_ and SENSOR#2-a_wz_

This paragraph focuses on the comparison between the frequency-weighted vertical acceleration values based on data collected from SENSOR#1 and SENSOR#2 respectively (Figure 14).

The dispersion of data points around the regression line and a non-unit slope agree that the two identical prototypes were placed in different positions inside the vehicle, which in turn differed, in the physical and mechanical characteristics and also in the recorded speeds: these factors significantly influence the final values of *a*_wz_ index.

### 5.3. Comparison between a_wz_ vs. IRI and a_wz_ vs. RN

For the purpose of this research, it is also important to consider the comparison between *a*_wz_ index, calculated from data collected in the field tests respectively by SENSOR#1 and SENSOR#2, and the values of IRI and RN related to the data collected by “Cartesio” (Figure 15).

As shown in Figure 15, the calculated *a_wz_* by the proposed system and the indices determined by RSP are strongly correlated in all cases with the coefficient of determination more than 0.83. Considering the *a_wz_* determined with acceleration data measured by IMU located in the same vehicle where the RN were measured, a higher coefficient of determination was obtained.

In summary, the results show very good correlations between the frequency-weighted vertical accelerations obtained from the proposed IMU-based devices and the ones from the LandMark. Moreover, the correlations between *a_wz_* and traditional pavement indices reveal that these low-cost devices can be regarded as reasonably reliable tools to assess the pavements decay in road networks with cost and difficulty of operation remarkably lower than traditional techniques. However, from the comparison between the two proposed sensors placed in different vehicles, it can be observed that the repeatability of the results depends on the speed and the physical-mechanical characteristics of the vehicle.

## 6. Conclusions

This work aimed to verify the feasibility of using a Raspberry-based IMU device to monitor the road pavement condition in urban areas. Tests were carried out using two identical Raspberry-based prototypes along about 21 km of urban and nonurban roads with flexible pavement. The validation test was performed employing concomitant measurements using the IMU LandMark10 + GPS and the “Cartesio” Road Asset Collection System vehicle.

Considering the comfort index *a_wz_* in accordance with ISO 2631 standard, the results showed very good correlations between the frequency-weighted vertical accelerations calculated with the proposed IMU (SENSOR#1) and the ones in the reference IMU. The coefficient of determination (R^2^ = 0.98) indicates a strong relation between measurements from the SENSOR#1 and the LandMark10, although the former presented a smaller accuracy and an *a_wz_* index value 10 per cent greater than the reference IMU. Besides, the comparison between the two Raspberry-based devices yielded a coefficient of determination (R^2^ = 0.73). This discrepancy is a consequence to the sensors were installed in different vehicles and in different positions inside each vehicle. This leads to an initial indication of how speed and the vehicle’s physical and mechanical characteristics may affect the estimation of the comfort indicator *a_wz_*.

Furthermore, the evaluation of the correlation between data gathered by “Cartesio” (IRI and RN) and the *a_wz_* indices calculated by SENSOR#1 and SENSOR#2 revealed a good consistency between the measurements. The correlation coefficients were in all arrangements greater than 0.82, implying a high correlation between the reference data and the measurements obtained from the proposed devices. It must be emphasized that the greatest correlation (R² = 0.90) was, as expected, verified between RN and the frequency-weighted vertical accelerations calculated from the SENSOR#2 located inside “Cartesio”.

It may be concluded that the proposed sensor can be considered a valuable tool for a quick, very low-cost road survey if considered that the repeatability of the results is conditioned by the speed and physical-mechanical characteristics of the vehicle. The proposed study is not intended to establish the Raspberry platform on the same level as the other precision devices. Instead, the correct interpretation is to provide an affordable tool that does not require dedicated staff and that can be easily installed in public service vehicles, local public transport vehicles, and even two-wheeled vehicles, widening the range of monitorable pavements (including sidewalks and bike lanes).

Since the described device is a prototype, it could be possible to perform improvements such as its integration with a GSM unit to transmit data directly to a server. In this context, for reasons of repeatability, information such as type of vehicle, position inside the vehicle, the fixing system and, finally, the speed at which the recording was carried out would be mandatory to enable a weighted evaluation of the measurements. Remaining within the scope of instrumentation refinement, it could be envisaged to develop a GIS system for the positioning and cataloguing of measurements in terms of *a_wz_* index in order to enable better integration with traditional measurement systems.

## Figures and Tables

**Figure 1 sensors-21-03127-f001:**
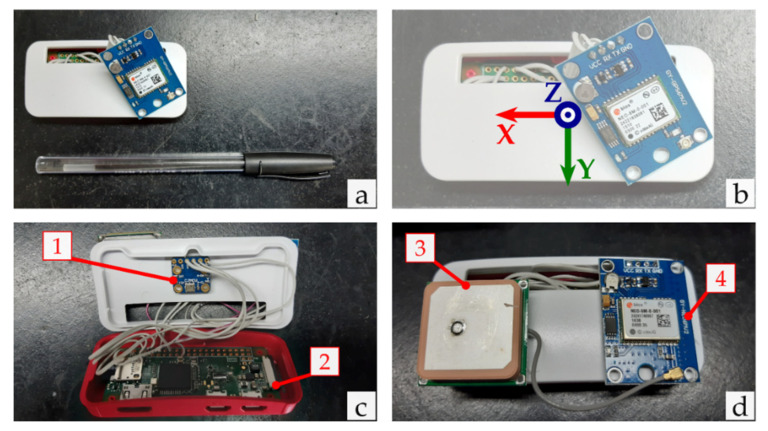
The core components of the developed device: (**a**) size comparison of the Raspberry + IMU module + GPS module assembly; (**b**) sensor axes orientation; (**c**) device with its case open and with a view of the IMU module (1) and the Raspberry Pi Zero W (2); (**d**) closed case with a view of the GPS antenna (3) and the u-Blox GPS module (4).

**Figure 2 sensors-21-03127-f002:**
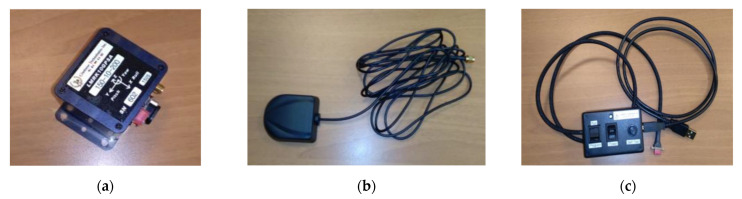
LandMark 10 GPSA-150-10-200 main component parts: (**a**) IMU; (**b**) the integrated GPS receiver; (**c**) the power supply.

**Figure 3 sensors-21-03127-f003:**
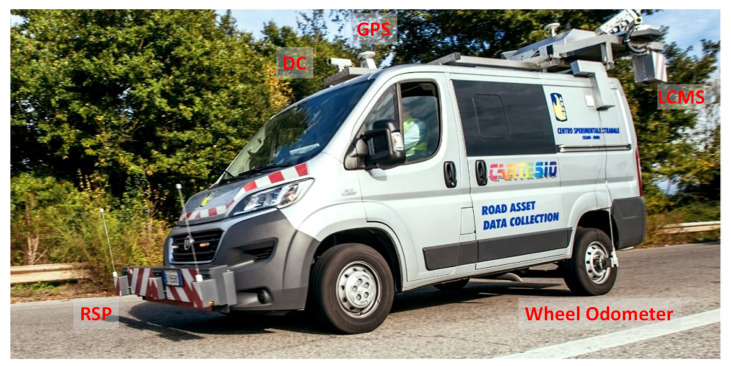
RACS employed for validation and its main components.

**Figure 4 sensors-21-03127-f004:**
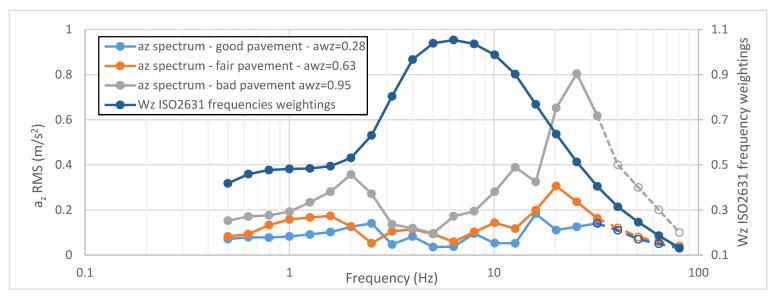
Three different spectra calculated starting from acceleration and ISO2631 frequencies weightings curve.

**Figure 5 sensors-21-03127-f005:**
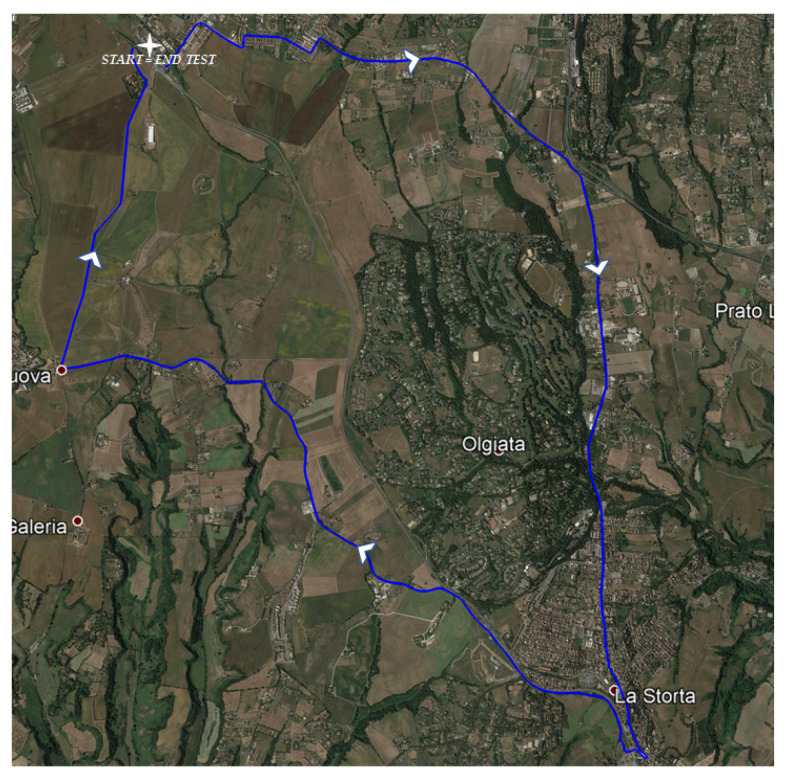
Examined roads and travel direction.

**Figure 6 sensors-21-03127-f006:**
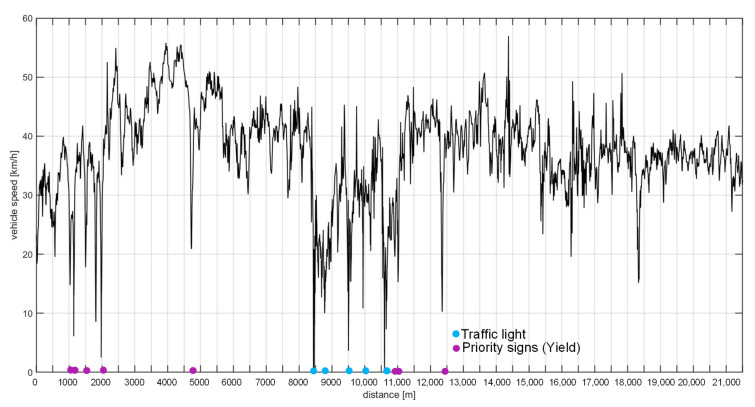
Recorded speed in a test vehicle.

**Figure 7 sensors-21-03127-f007:**
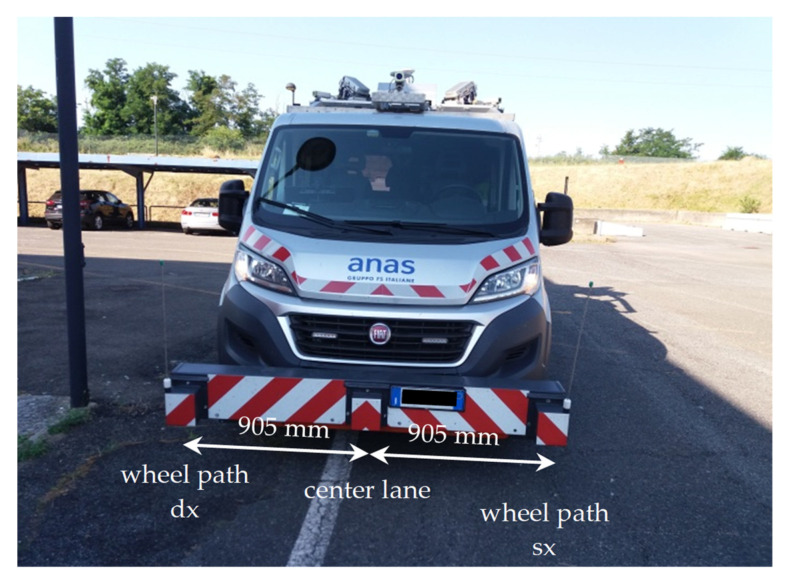
“Cartesio” front view; the RSPs are assembled on a bar in front of the vehicle.

**Figure 8 sensors-21-03127-f008:**
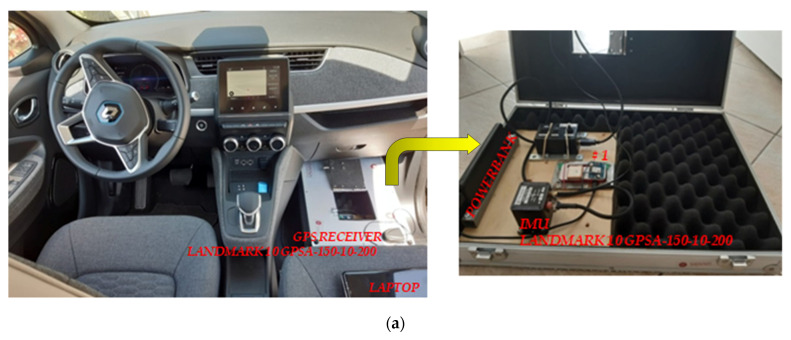
Position of the prototypes: (**a**) SENSOR#1 inside Renault Zoe; (**b**) SENSOR#2 inside Cartesio.

**Figure 9 sensors-21-03127-f009:**
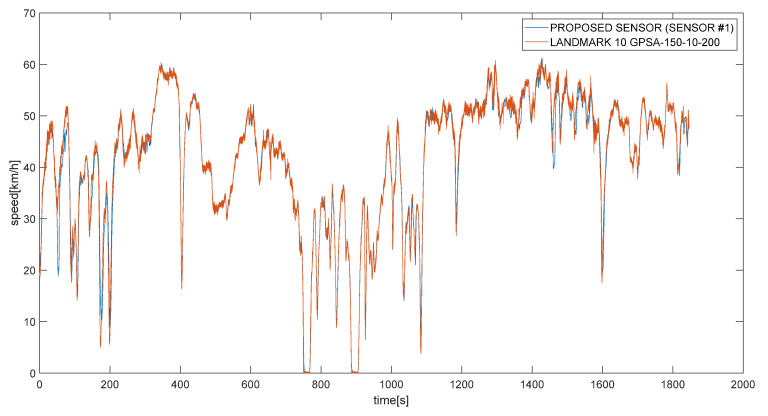
Recorded speed in a test from different instruments placed in the same vehicle: the proposed sensor #1 (blue) and the LandMark 10 GPSA-150-10-200 (orange).

**Figure 10 sensors-21-03127-f010:**
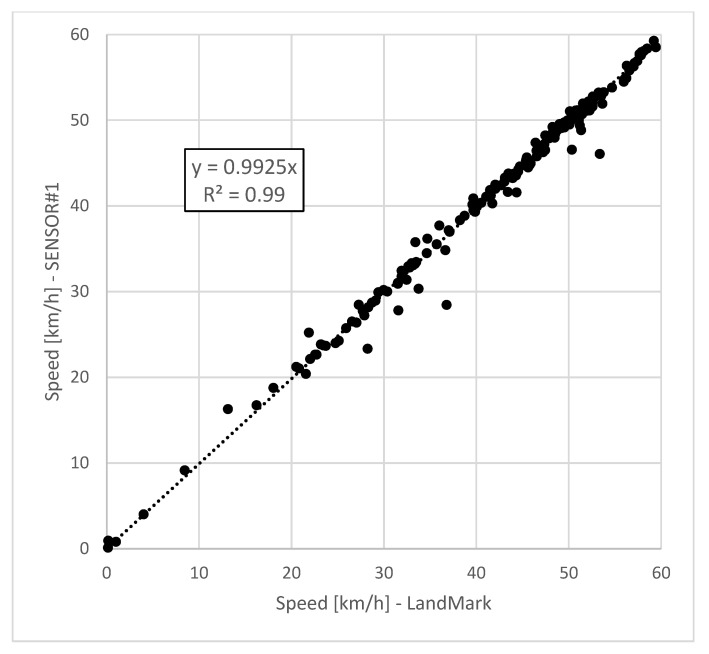
Comparison between speed values collected by LandMark 10 GPSA-150-10-200 and SENSOR#1 at the same time sample.

**Figure 11 sensors-21-03127-f011:**
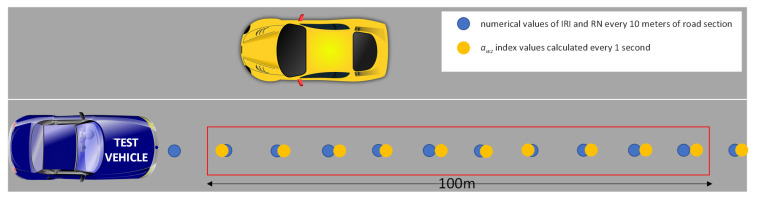
Scheme of the measurement positions along the sub-section of road.

**Figure 12 sensors-21-03127-f012:**
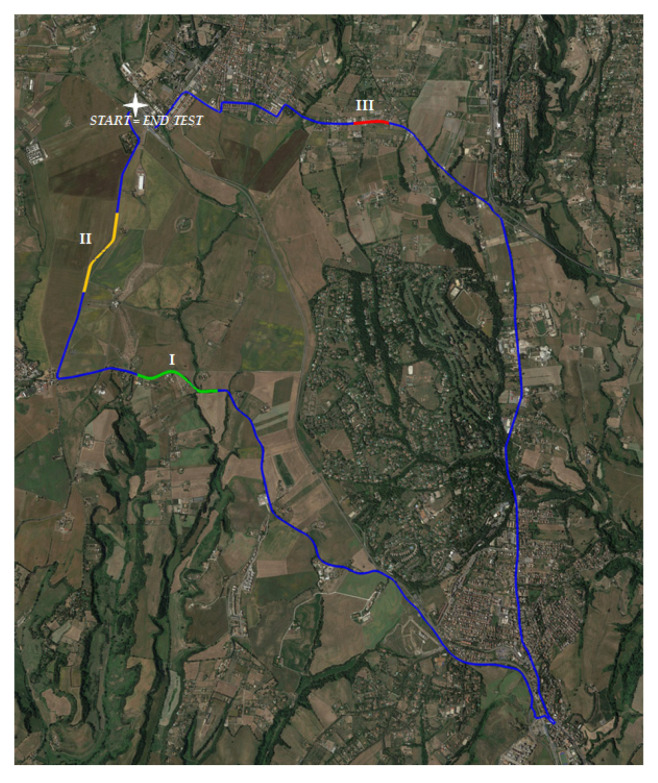
Examined sections for each pavement condition category.

**Figure 13 sensors-21-03127-f013:**
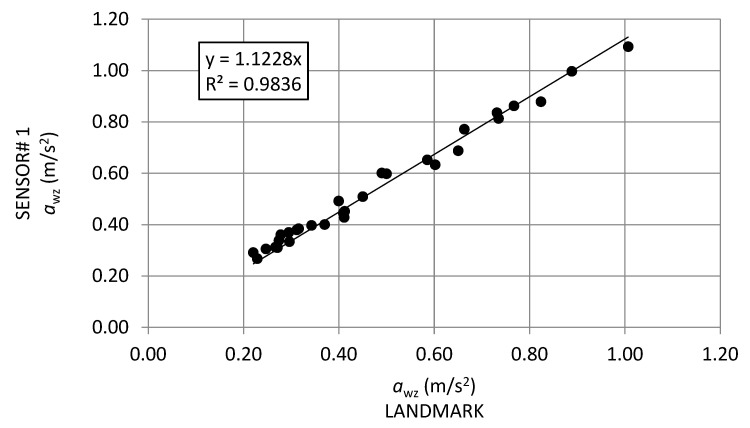
Linear regression between SENSOR#1-*a_wz_* vs. LANDMARK-*a_wz_.*

**Figure 14 sensors-21-03127-f014:**
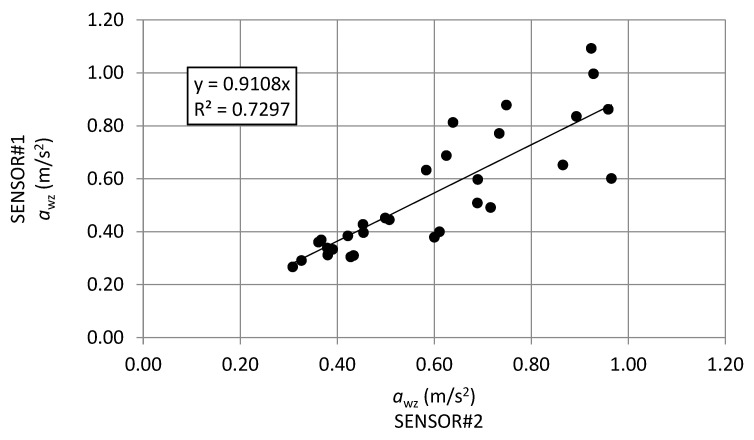
Linear regression SENSOR#1-*a_wz_* vs. SENSOR#2-*a_wz_.*

**Figure 15 sensors-21-03127-f015:**
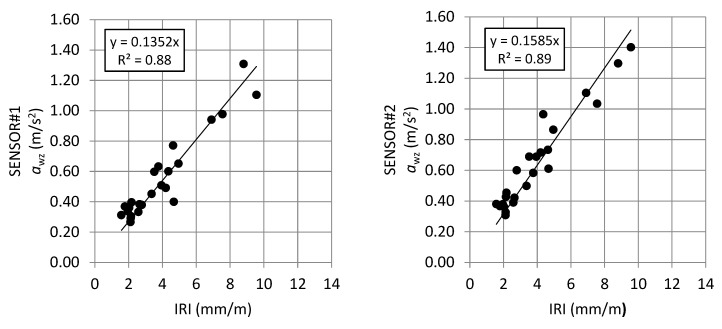
Linear regression *a*_wz_–IRI vs. *a*_wz_–RN.

**Table 1 sensors-21-03127-t001:** MPU-9250 accelerometer, gyroscope, and magnetometer main features [51].

Property	Accelerometer	Gyroscope	Magnetometer
Full-scale range	User-programmable: +− 2, 4, 8 or 16 g	User-programmable: 250, 500, 1000 or 2000 °/s	+− 4800 µT
Noise spectral density	300 µg/√Hz	0.01 °/s/√Hz	-
Sensitivity scale factor	User programmable: 16,384, 8192, 4096 or 2048 LBS/g	User-programmable: 131, 65.5, 32.8 or 16.4 LBS/(°/s))	0.6 µT/LSB
Sample rate	up to 4000 Hz	up to 8000 Hz	up to 8 Hz

**Table 2 sensors-21-03127-t002:** U-blox NEO-6M mini GPS module main features [56].

SBAS	Wide Area Augmentation System, European Geostationary Navigation Overlay Service, and Multi-functional Satellite Augmentation System
Maximum update rate	5 Hz
Time-To-First-Fix ^1^	Cold or warm start: 27 s Hot start: 2 s Aided start: < 3 s
Horizontal position error ^2^	GPS: 2.5 m
	SBAS: 2.0 m
Velocity error ^2^	0.1 m/s
Bearing error ^2^	0.5 degree

^1^ Satellites at −130 dBm. ^2^ Circular Error Probability (CEP) 50%, satellites at −130 dBm, obtained from 24-h static position solution.

**Table 3 sensors-21-03127-t003:** Comfort levels related to *a_wz_* threshold values as proposed by ISO 2631 for public transport.

$a_{wz}\;\mathrm{Values}\;(m/s^{2})$	Ride Number
less than 0.315	Not uncomfortable
0.315–0.63	Little uncomfortable
0.5–1.0	Fairly uncomfortable
0.8–1.6	Uncomfortable
1.25–2.5	Very uncomfortable
more than 2.5	Extremely uncomfortable

**Table 4 sensors-21-03127-t004:** Ride Number Thresholds

Description	Ride Number
Perfect	5.0
Very Good	4.5
	4.0
Good	3.5
	3.0
Fair	2.5
	2.0
Poor	1.5
	1.0
Very poor	0.5
Impassable	0.0

**Table 5 sensors-21-03127-t005:** Characteristics of the roads for system validation.

Branch	Length (m)	Speed Limit (km/h)	Road Classification	Traffic Light (Number)	Priority Road Signs (Number)
A	500	50	Urban	NO	NO
B	550	30	Urban	NO	NO
C	100	30	Urban	NO	YES (2)
D	650	30	Urban	NO	YES (1)
E	180	30	Urban	NO	YES (1)
F	2800	50	Urban	NO	YES (1)
G	5700	50	Nonurban	YES (5)	NO
H	230	30	Urban	NO	YES (2)
I	7400	50	Nonurban	NO	YES (1)
A	3600	50	Urban	NO	NO

**Table 7 sensors-21-03127-t007:** Pavement condition category associated to the roughness thresholds considered in this paper.

Pavement Condition Category	IRI [mm/m]
GOOD	IRI < 3
FAIR	3–5
POOR	IRI > 5

**Table 8 sensors-21-03127-t008:** Characteristics of examined sections.

Section Name	Condition	Chainage (km)		Length (m)	Number of Sub-Sections ^1^
		Start	End		
I	Good	16 + 300	17 + 300	1000	10
II	Fair	19 + 200	20 + 200	1000	10
III	Poor	2 + 800	3 + 200	400	4
				2400	24

^1^ It was calculated the average value of each index per sub-section length of 100 m.

## Data Availability

Not applicable.

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
