# Peer review of "Validation of a Low-Cost Pavement Monitoring Inertial-Based System for Urban Road Networks"

_sensors, 2021, doi:10.3390/s21093127_

Round 1

Reviewer 1 Report

This study validates the effectiveness of a low-cost fast pavement monitoring inertial sensor (i.e., the Inertial Measurement Unit (IMU)) in evaluating the road condition of urban road network, through comparing the evaluation results using the proposed IMU with that from a reference IMU. The topic of this manuscript fits in well the scope of sensors journal. The paper quality needs further improvement regarding the writing quality, the method design, and the result presentation and discussions. Therefore, major revision is recommended before it can be further considered for publication.

I.    Academic writing: The writing quality of this manuscript is generally below the standard for journal articles. The structures of sentences and paragraphs are not clear, and there are many grammatical errors. Generally, the lack of writing quality largely hinders the readers from understanding the technical part clearly. The authors would like to rewrite this paper using more concise and technical expressions instead of verbose and vague ones, since this manuscript is supposed to be published as a technical paper on the Sensors journal. Please see below for some examples and consider scrutinizing the academic writing of the whole manuscript.
1.    Line 14: If the IMU presented in this manuscript is newly proposed, you may want to say a “novel” IMU…. Or you would like to give it a new name, such that it can be easily distinguished from other IMUs you referred to later in the context.
2.    Line 19: IMU should be defined when it first appears.
3.    Line 18: The structure of the sentence “Considering ….” is unclear. Please consider breaking it into two short and concise sentences.
4.    Line 23: Check the grammar “The results were shown….”
5.    Line 24: Check the grammar “…were showed….”
6.    Line 27: Please consider using academic expressions, such as “expensive” instead of “not cheap”. 
7.    Line 40: “require”.
8.    Line 57: “to depend on the length of the profile”.
9.    Line 60: “so to accept higher IRI thresholds for the roads where the 60 design speed is lower.”
10.    Line 87: The structure of this paragraph is messy.
11.    Please consider combining the three paragraphs from Line 81 to Line 92.
12.    Line 93: What do you mean by “To the shortcomings highlighted”? Please consider breaking this long sentence to several short and concise sentences for clarity.
13.    Line 107: What sensors do you mean by “These sensors”? No sensor is mentioned in the paragraph above.
14.    Line 108: “In fact,….” This sentence is too long and hard to follow. Please write concisely.  Please double check with the editor whether this is the correct style of writing a technical paper to be published on the Sensors journal.
II.    Methodology and Results
1.    In Section 2, I didn’t see any sentence regarding the method proposed in this study. It is more like a review of pavement evaluation methods. So, it looks like the literature review does not end until Line 234 on Page 6.
2.    The results of comparison are very good, as shown in the figures. However, the authors need to highlight or clarify the merits of developed sensing system compared with the state-of-the-art ones.
3.    Further review will be conducted after the writing quality is further improved.

Author Response

Reviewer #1:

This study validates the effectiveness of a low-cost fast pavement monitoring inertial sensor (i.e., the Inertial Measurement Unit (IMU)) in evaluating the road condition of urban road network, through comparing the evaluation results using the proposed IMU with that from a reference IMU. The topic of this manuscript fits in well the scope of sensors journal. The paper quality needs further improvement regarding the writing quality, the method design, and the result presentation and discussions. Therefore, major revision is recommended before it can be further considered for publication.

  1. Academic writing: The writing quality of this manuscript is generally below the standard for journal articles. The structures of sentences and paragraphs are not clear, and there are many grammatical errors. Generally, the lack of writing quality largely hinders the readers from understanding the technical part clearly. The authors would like to rewrite this paper using more concise and technical expressions instead of verbose and vague ones, since this manuscript is supposed to be published as a technical paper on the Sensors journal. Please see below for some examples and consider scrutinizing the academic writing of the whole manuscript.

                [Answer] Thank you for your remarks. Besides the improvements listed in the items below; we have made the following changes to the text:

  • “monitor the road condition indirectly” instead of “monitor indirectly the road condition” (lines 17 and 133);
  • “the process interests on” instead of “the process interests on” (line 35);
  • “North America” instead of “the North America” (line 43);
  • “problems that limit” instead of “problems that limits” (line 68);
  • “alternative systems that can” instead of “alternative systems than can” (line 77);
  • “methods involving” instead of “methods involve” (line 79);
  • “these difficulties” instead of “these difficult” (line 111);
  • “validation measurements” instead of “validation measures” (137);
  • “conversion of values from” instead “conversion of values form” (line 287);
  • removal of “positioning” (line 301);
  • correction of “preliminary” and “comparison”, which were hyphenated (lines 316 and 317);
  • “interpolation of position, velocity, and time (PVT)” instead of “interpolation of PVT” (line 317);
  • “components” instead of “component parts” (line 347); 
  • “allows eliminating” instead of “allows to eliminate” (line 563);
  • “gave us the opportunity” instead of “allowed us” (367);
  • “Figure” instead of “the Figure” (line 397);
  • “the Italian […] Department” instead of “Italian […] Department” (line 377);
  • “Figure” instead of “the Figure” (line 428);
  • “spectra” (plural) instead of “spectrum” (line 429);
  • “is” instead of “are” (line 429);
  • “three different spectra” (plural) instead of “three different spectrum” (line 432);
  • “Figure” instead of “the Figure” (line 434);
  • “is regarded as a standard” instead of “is regarded to as a standard” (line 438);
  • “one of the most adopted indexes” instead of “one of the most adopted index” (line 453);
  • “road safety” instead of “the road safety” (line 456);
  • “a computer program that implements” instead of “computer program that implementing” (line 460);
  • “Equation” instead of “the Equation” (line 461);
  • “depend on” instead of “depends from” (line 469);
  • “also lengths of 50 m and 20 m” instead of “also length of 50 m and 20 m” (line 474);
  • “contribution of the single event bumps with respect to” instead of “contribution of the single event bumps respect to” (line 475);
  • “is the result” instead of “is result” (line 478);
  • “pavement roughness” instead of “roughness pavement” (line 482);
  • “obtained by determining” instead of “obtained determining” (line 487);
  • “linked to subjective opinions” instead of “linked to subjective opinion” (line 488);
  • “it is possible to evaluate” instead of “it is possible evaluate” (line 488);
  • “can be found” instead of “can be find” (line 497);
  • “an overlap […] was” instead of “an overlap […] were” (line 510);
  • “in Table 6” instead of “in the Table 6” (line 542);
  • “was obtained” instead of “was resulted” (line 567);
  • “as a result” instead of “as result” (line 572);
  • removal of “consequently” (line 577);
  • “constant long road sub-sections” instead of “constant long sub-sections of road” (line 581);
  • “conditions that did not always allow” instead of “conditions such as not to always allow” (line 588);
  • “it is important” instead of “it’s important” (line 616)

  1. Line 14: If the IMU presented in this manuscript is newly proposed, you may want to say a “novel” IMU…. Or you would like to give it a new name, such that it can be easily distinguished from other IMUs you referred to later in the context.

                [Answer] We rewrote this excerpt and defined our device as a “novel inertial sensor-based system” (line 14). Moreover, the sensors are named “device” or “sensor#1” and “sensor#2” throughout the text considering the two different prototypes used. The other reference high precision IMU + GPS module used for the validation is named “LandMark” , the same abbreviated name assigned by the seller.

  1. Line 19: IMU should be defined when it first appears.

                [Answer] We added this definition in line 115.

  1. Line 18: The structure of the sentence “Considering ….” is unclear. Please consider breaking it into two short and concise sentences.

                [Answer] We rewrote this excerpt (line 19): “Considering 21 km of roads with different levels of pavement decay, validation measurements were performed using the novel sensor, a high-performance inertial based navigation sensor, and a Road Surface Profiler.”

  1. Line 23: Check the grammar “The results were shown….”
  2. Line 24: Check the grammar “…were showed….”

                [Answer] For both, we corrected the grammar (lines 24 and 26)

  1. Line 27: Please consider using academic expressions, such as “expensive” instead of “not cheap”.

                [Answer] We replaced “not cheap” by “expensive” (line 28).

  1. Line 40: “require”.

                [Answer] We replaced “require” by “requires” (line 41).

  1. Line 57: “to depend on the length of the profile”.

                [Answer] Excerpt replaced by “depending on the profile length” (line 58).

  1. Line 60: “so to accept higher IRI thresholds for the roads where the design speed is lower.”

[Answer] Excerpt replaced by (line 61): “operating speed of the road [22,23], so as to accept higher IRI thresholds for the roads where the operating speed is lower”

  1. Line 87: The structure of this paragraph is messy.

[Answer] We rewrote this paragraph (line 90): “For this reason, smartphones were recently proposed to evaluate road condition over the world with different approaches: some apps try to estimate IRI along the surveyed road (divided into constant segments, 20-50-100 m) so as to provide a typical evaluation of pavement quality [40]. Other apps propose new indexes [41] or categorize the acceleration peaks [42] in order to evaluate pavement conditions.”

  1. Please consider combining the three paragraphs from Line 81 to Line 92.

                [Answer] The paragraphs were combined (space removed in lines 86 and 89)

  1. Line 93: What do you mean by “To the shortcomings highlighted”? Please consider breaking this long sentence to several short and concise sentences for clarity.

                [Answer] We rewrote this sentence (line 94): “The urban pavements present also other management problems that consist in the inadequacy of thresholds for the roughness indicators currently in use considering the operating speeds, generally below 50 km/h [22,43,44].”

  1. Line 107: What sensors do you mean by “These sensors”? No sensor is mentioned in the paragraph above.

                [Answer] In line 109, we redefined the “pavement evaluation systems”, which are the systems we refer to in the line 113 (“these sensors” replace by “these systems).

  1. Line 108: “In fact,….” This sentence is too long and hard to follow. Please write concisely. Please double check with the editor whether this is the correct style of writing a technical paper to be published on the Sensors journal.

                [Answer] We rewrote this sentence (line 107): “Generally, they [managers] choose maintenance strategies and the related interventions to be taken regardless of any monitoring activity (time-based maintenance) or in consequence of failures (run-to-failure maintenance) with serious losses in terms of direct and indirect costs for the community”.

  1. Methodology and Results
  2. In Section 2, I didn’t see any sentence regarding the method proposed in this study. It is more like a review of pavement evaluation methods. So, it looks like the literature review does not end until Line 234 on Page 6.

                [Answer] We defined new sections. Sections 2 describes the pavement monitoring instruments used in our research, comprising the description of the proposed sensors (subsection 2.1) and the other tools used as references (2.2 for LandMark IMU+GPS and 2.3 for the road asset collection system). Section 3 describes the pavement evaluation indices used in our research, comprising the weighted vertical acceleration (calculated from the proposed sensor and from the one of reference, LandMark) and the indices (IRI and RN) calculated by the road asset collection system. 

  1. The results of comparison are very good, as shown in the figures. However, the authors need to highlight or clarify the merits of developed sensing system compared with the state-of-the-art ones.

[Answer] We added the following paragraph in line 526, at the end of “Results and Discussion” section: “In summary, the results show very good correlations between the frequency-weighted vertical accelerations obtained from the proposed IMU-based devices and the ones from the LandMark. Moreover, the correlations between awz and traditional pavement indexes reveal that these low-cost devices can be regarded as reasonably reliable tools to assess the pavements decay in road networks with cost and difficulty of operation remarkably lower than traditional techniques. However, from the comparison between the two proposed sensors placed in different vehicles, it can be observed that the repeatability of the results depends on the speed and the physical-mechanical characteristics of the vehicle.”

3.    Further review will be conducted after the writing quality is further improved.

Reviewer 2 Report

The paper deals with the use of a low cost IMU-GPS system for the assessment of the road pavement condition.
The problem is really well introduced with a clear discussion of the present problems also reporting the current international standards.

The section with the description of the electronic system is rather inaccurate and should be improved.
The experimental results reported in the paper are relative to the proposed system compared with a similar commercial IMU-GPS device but no comparison is reported about the state of the measured pavements that could be acquired by using the Cartesio Lab system.
It seems that the Cartesio Lab system has been used simply as a further vehicle without make any use of the instrumentation (well described in the paper) onboard on it.

Some minor issues

Line 20 and 22 Acronyms should not be used neither defined in the abstract
Line 149 the number of open brackets seems do not match with the closed ones
Line 161 the sentence "fs = 1⁄Δ? in Hz, where Δ? is the signal sampling" is trivial
Line 249 "re-mote" ??
Line 249 the authors state that the wireless control of the Raspberry board avoids "the need for uninstallation and reinstallation" 
         It is not clear what has to be uninstalled and reinstalled. The microprocessor board once programmed does not need anything more and the uploaded program runs freely on it. 
         the data download is not usually referred as 
Line 263 The MPU-9250 does not have 10 DoF but it is a 9-DoF only. Some of the modules available on the market have the pressure module BMP280
         onboard.
Table 3     The values reported in the "Output data rate" row, actually are the maximum sample rate of the three sensors (Accelerometer Gyro 
         and Magnetometer). The output data rate is usually related as the speed of the ports of the microprocessor that is different from the sample rate.
Line 283 Output data rate --> sample rate
line 305 "com-parison" ??
line 327 No reference has been reported on the LandMark 10 GPSA-150-10-200 system. At least the manufacturer website or a datasheet
         should be reported
Line 339 No reference has been reported on the software GLAMR
Line 353 No reference has been reported on the mobile laboratory named “Cartesio (at least the ANAS website...)
Lines 397-404 the same data have been reported above (lines 347-397)
Line 425 is the speed computed by integration of the data from the IMU or simply read from GPS device?

Author Response

Reviewer #2:

The paper deals with the use of a low cost IMU-GPS system for the assessment of the road pavement condition.

The problem is really well introduced with a clear discussion of the present problems also reporting the current international standards.

[Answer] Thank you for your remarks.

The section with the description of the electronic system is rather inaccurate and should be improved.

                [Answer]: The section is improved considering some changes in grammatical form and rewriting some phrases.

The experimental results reported in the paper are relative to the proposed system compared with a similar  commercial IMU-GPS device but no comparison is reported about the state of the measured pavements that could be acquired by using the Cartesio Lab system. It seems that the Cartesio Lab system has been used simply as a further vehicle without make any use of the instrumentation (well described in the paper) onboard on it.

                [Answer] We rewrote the Cartesio description (section 2.3, line 360). In lines 531, 567, and 635 we emphasized that Cartesio yields IRI and RN indices.

Some minor issues

Line 20 and 22 Acronyms should not be used neither defined in the abstract

                [Answer] We removed the definition of the acronyms in the abstract.

Line 149 the number of open brackets seems do not match with the closed ones

                [Answer] We removed the extra bracket (line 412)

Line 161 the sentence "fs = 1⁄Δ? in Hz, where Δ? is the signal sampling" is trivial

                [Answer] We removed this formula and replaced “output data rate” by “sampling frequency” (line 292).

Line 249 "re-mote" ??

                [Answer] We tried to correct this mistake and the ones similar that are consequences of MS Word automatic hyphenation.

Line 249 the authors state that the wireless control of the Raspberry board avoids "the need for uninstallation and reinstallation" It is not clear what has to be uninstalled and reinstalled. The microprocessor board once programmed does not need anything more and the uploaded program runs freely on it.  The data download is not usually referred as [sic]

                [Answer] The statement wanted to discuss the physical installation of the devices and the lack of need to remove them from their cases or their positions in a vehicle. In order to clarify this fact, we added the word “physical” to characterize the re/uninstallation.

Line 263 The MPU-9250 does not have 10 DoF but it is a 9-DoF only. Some of the modules available on the market have the pressure module BMP280 onboard.

                [Answer] In fact, the MPU-9250 used in this research does have a barometer (the pressure module BMP280).

Table 3     The values reported in the "Output data rate" row, actually are the maximum sample rate of the three sensors (Accelerometer Gyro  and Magnetometer). The output data rate is usually related as the speed of the ports of the microprocessor that is different from the sample rate.

Line 283 Output data rate --> sample rate

                [Answer] For Table 1, line 285 (former Table 3), and for line 296, we replaced “output data rate” for “sample rate”. Moreover, we used “update rate” in line 315 as a more accurate term for GPS. 

line 305 "com-parison" ??

                [Answer] We corrected this mistake.

line 327 No reference has been reported on the LandMark 10 GPSA-150-10-200 system. At least the manufacturer website or a datasheet should be reported

                [Answer] We added the reference number [61] - GLADIATOR TECHNOLOGIES Web page, available at https://gladiatortechnologies.com/

Line 339 No reference has been reported on the software GLAMR

[Answer] We added the reference number [63] - GLAMR Software Web page, available at https://gladiatortechnologies.com/software-development-kit/.

Line 353 No reference has been reported on the mobile laboratory named “Cartesio (at least the ANAS website...)

[Answer] We added the references number [64] (ANAS Web site, available at https://www.stradeanas.it/it) and 64 –(FS GROUP web site), in line 366.

Lines 397-404 the same data have been reported above (lines 347-397)

                [Answer] In these topics (lines 529-537), we described critical features of these instruments (sample rate, post-processing issues, length for IRI calculation) regarding their use as references.

Line 425 is the speed computed by integration of the data from the IMU or simply read from GPS device?

                [Answer] In line 557, we added the speeds were gathered from GPS receivers.

Round 2

Reviewer 1 Report

This authors have made considerable revisions to improve the overall quality of this manuscript. It can be accepted as it is. The authors may want to consider citing the following works in their manuscript.

Wambold, J. C., L. E. Defrain, R. R. Hegmon, K. Macghee, J. Reichert, and E. B. Spangler. "State of the art of measurement and analysis of road roughness." Transportation research record 836 (1981): 21-29.

Zhang, Zhiming, Fodan Deng, Ying Huang, and Raj Bridgelall. "Road roughness evaluation using in-pavement strain sensors." Smart Materials and Structures 24, no. 11 (2015): 115029.

Kropac, Oldrich, and Peter Mucka. "Indicators of longitudinal unevenness of roads in the USA." International journal of vehicle design 46, no. 4 (2008): 393-415.

Zhang, Zhiming, Chao Sun, Raj Bridgelall, and Mingxuan Sun. "Application of a machine learning method to evaluate road roughness from connected vehicles." Journal of Transportation Engineering, Part B: Pavements 144, no. 4 (2018): 04018043.

Múčka, Peter. "Longitudinal road profile spectrum approximation by split straight lines." Journal of transportation engineering 138, no. 2 (2012): 243-251.

Reviewer 2 Report

The paper has been suitable improved following the reviewers' suggestions.
I think that it can be accepted for publication on Sensors.